# Enhancement of oral bioavailability of ibrutinib using a liposil nanohybrid delivery system

**Fareeaa Ashar**[1], **Asif Ansari Shaik Mohammed**[2], **S. Selvamuthukumar**[1]*

1 Research Scholar, Department of Pharmacy, Annamalai University, Annamalainagar, Chidambaram, Tamil Nadu, India, 2 Department of Clinical Pharmacy, College of Pharmacy, King Khalid University, Abha, Kingdom of Saudi Arabia

* smk1976@gmail.com

**Data Availability Statement:** All relevant data are within the manuscript.

**Funding:** The authors extend their appreciation to the Deanship of Scientific Research at King Khalid University for funding this work through Large

## Abstract

Liposils, synthesized via the liposome templating method, offer a promising strategy for enhancing liposome stability by employing a silica coating. This study focuses on the development of nanocarriers utilizing silica-coated nanoliposomes for encapsulating the poorly water-soluble drug, ibrutinib. Ibrutinib-loaded nanoliposomes were meticulously formulated using the reverse-phase evaporation technique, serving as templates for silica coating, resulting in spherical liposils with an average size of approximately 240 nanometers. Comprehensive characterization of the liposil's physical and chemical properties was conducted using various analytical methods, including dynamic light scattering, transmission electron microscopy, Fourier-transform infrared spectroscopy, and X-ray diffraction analysis. Liposils demonstrated superior performance compared to ibrutinib-loaded nanoliposomes, showing sustained drug release profiles in simulated intestinal fluids and resistance to simulated gastric fluid, as confirmed by dissolution studies. Moreover, ibrutinib liposils exhibited a significant increase in half-life (4.08-fold) and notable improvement in bioavailability (3.12-fold) compared to ibrutinib suspensions, as determined by pharmacokinetic studies in rats. These findings underscore the potential of liposils as nanocarriers for orally delivering poorly water-soluble drugs, offering enhanced stability and controlled release profiles, thereby improving bioavailability prospects and therapeutic efficacy. This approach holds promise for addressing challenges associated with the oral administration of drugs with limited solubility, thereby advancing drug delivery technologies and clinical outcomes in pharmaceutical research and development.

## 1. Introduction

In aqueous environments, phospholipids spontaneously assemble into vesicular structures known as liposomes, characterized by bilayer comprising phospholipids [1]. These liposomes have garnered extensive research attention due to their unique properties, which encompass their capacity to shield drug molecules from enzymatic degradation, prolong systemic

Group Research Project under grant number
RGP2/54/44.

**Competing interests:** The authors have declared
that no competing interests exist.

circulation, enhance bioavailability, enable targeted drug delivery, mitigate side effects and toxicity, and offer ease of manufacturing [2]. Liposomes have been the subject of meticulous investigation for their role as effective drug carriers. They serve as versatile vehicles, accommodating a diverse range of drugs. Hydrophilic drugs find their place within the aqueous core of liposomes, while hydrophobic drugs become integrated into the lipid bilayer [3]. However, despite their promising attributes, liposomes have not fully realized their potential due to challenges associated with *in vivo* stability and limited applicability in environments with acidity levels approaching neutrality. Several mechanisms of destabilization, including creaming, fusion, and vesicle aggregation, pose substantial challenges by increasing the size of liposomal vesicles during both production and storage [4]. Consequently, the pursuit of maintaining consistent vesicle sizes over time remains a desirable objective in liposome research.

To enhance the liposomes stability and protect their fragile structure several systems have been proposed, such as Polymerosomes [5] Responsive molecular gates embedded in self-assembled inorganic and organic shells [6], Liposomes with a silica nanoparticle coating [7], Silica layer-liposils [8], Lyophilosomes [9], Polymer caged liposomes [10], PEGylated liposomes [11], Layerosomes-polyelectrolyte layer-by-layer assembly [12], Colloidosomes [13] and polycationic brushes [14]. However, the practical implementation of these techniques has been limited due to inconsistent necessities necessary for both silica biomineralization and liposome stabilization at the preparative stage. Recent investigations have highlighted the feasibility of creating hollow silica nano-shell systems by employing nanoscale liposomes as templates for silica deposition [15]. One notable outcome of this approach is the development of Liposil, a silica-coated liposome achieved through a sol-gel reaction using liposome templating. Liposil exhibits promise in the realm of drug release strategies and storage, particularly for oral administration. This promise is attributed to its robust silica shell, which remains stable at pH 1.2 and facilitates controlled drug release at pH 7.4. The utilization of sol-gel-derived silica holds appeal due to its biocompatibility, chemical inertness, and cost-effectiveness, rendering it a viable avenue for the formulation of oral liposomal drug delivery systems [16]. The silica coating serves a dual purpose: it shields the encapsulated liposomes from the effects of temperature and pH variations while preserving the distinctive properties of the lipid bilayer. Furthermore, this coating enables the incorporation of both hydrophobic and hydrophilic medications within the liposomes.

Ibrutinib, a weak base medication having a 3.74pKa, is an anticancer agent targeting B-cell malignancies [17]. Despite its solubility in dimethyl sulfoxide and methanol, it is practically insoluble in water; resulting in low oral bioavailability (2.9%) [18]. Ibrutinib's *in vivo* anticancer activity following oral administration is diminished by its pH-dependent solubility, which renders it almost insoluble at pH 3 to 8 but somewhat soluble at pH 1.2. Therefore, there is a need for an improved formulation with better bioavailability and higher efficacy. Various methods, such as co-solvency, crystalline changes, solid dispersion technology, and others, have been developed for this purpose [19–23]. These approaches have drawbacks, such as using expensive, specialized excipients with scalability challenges, and none have been reported to improve drug stability.

The primary aim of this investigation was to scrutinize the in vivo and in vitro efficacy of silica-coated nanoliposomes encapsulating ibrutinib. Through a meticulous evaluation encompassing oral bioavailability, we conducted a comprehensive comparative analysis juxtaposing liposils, nanoliposomes, and a suspension of ibrutinib. Our findings elucidated a remarkable enhancement in the oral bioavailability of the hydrophobic drug, ostensibly facilitated by the incorporation of a silica coating onto the liposomal surface. This discernible augmentation underscores the potential utility of tailored nanoliposomal formulations in enhancing the therapeutic efficacy and translational potential of water-insoluble pharmaceutical agents.

## 2. Materials and methods

### 2.1 Materials

Ibrutinib was brought from MSN laboratories Pvt. Ltd. Hyderabad, India. From Sigma Aldrich (India) procured Pepsin, Deuterium oxide, Dicetyl phosphate, Tetraethyl orthosilicate, Cholesterol, and L-α-Phosphatidylcholine—(From Soybean). Stearylamine, Sodium fluoride, Acetonitrile, Trifluoroacetic acid, Chloroform and diethyl ether were purchased from S.D. Fine Chemicals Ltd., Mumbai, India. Dialysis membrane (Sartorius cut off 12,000 Da) was procured from Hi-Media Laboratories, India. All chemicals and reagents used in the study were analytical grade, and Milli-Q water (Millipore) was used throughout.

### 2.2 Reverse phase high performance liquid chromatography (RP-HPLC)

Quantitative estimation of ibrutinib was performed using a Shimadzu RP-HPLC system located in Cort, Japan. The mobile phase consisted of acetonitrile and ultrapure water (adjusted to pH 5.0 using dilute trifluoroacetic acid). The acetonitrile percentage was dynamically adjusted at specified time intervals throughout the gradient elution process. Initially, it was set at 40% for the first minute, gradually increasing to 90% between 1 to 2 minutes, maintaining this composition from 2 to 4 minutes, and then decreasing back to 40% from 4 to 6 minutes. A UV-Visible detector (SPD-20A) operating at 258 nm was employed to monitor the changes in composition over the course of 6 to 10 minutes [24, 25].

### 2.3 Preparation of ibrutinib nanoliposomes

The preparation of ibrutinib nanoliposomes was accomplished utilizing the reverse-phase evaporation technique, as detailed in a prior publication [25]. Initially, a mixture of ibrutinib, cholesterol, phosphatidylcholine, and sterylamine was dissolved in a solution of chloroform and diethyl ether within a 50 ml round bottom flask. Subsequently, the solvents were removed via rotary evaporation at 200 rpm and 37 ˚C under reduced pressure. To eliminate any residual solvents, the sample was subjected to N2 gas purging for 10 minutes, followed by placement in a vacuum oven for 24 hours at room temperature. The hydrated thin film was then sonicated at 180 watts and subjected to magnetic stirring to achieve homogeneous nanoliposomes in a 30 ml phosphate buffer. The resulting water-in-oil emulsions were stored in a refrigerator at a temperature of 4˚C until further characterization could be conducted [25].

### 2.4 Preparation of ibrutinib liposils

Silica-coated liposomes were prepared using a modified sol-gel technique, as previously documented [25]. Initially, tetraethyl orthosilicate, an inorganic hydrophobic precursor, underwent hydrolysis in phosphate-buffered saline (PBS) at pH 7.4 and was agitated at 40˚C for duration of two days. The resulting silicate solution was introduced to the dispersion of ibrutinib nanoliposomes at room temperature, and gentle stirring at 200 rpm was maintained for 24 hours. Sodium fluoride, acting as a catalyst at a molar ratio of 4% to tetraethyl orthosilicate, was employed to facilitate the silica aging process. The reaction medium continued to be stirred at room temperature for an additional 48 hours. To eliminate any residual sodium fluoride and unreacted precursors, the resulting dispersion of ibrutinib liposils was dialyzed for one hour. The solid product was subsequently filtered through a 200 nm pore size filter and dried at 40˚C for 24 hours.

## 2.5 Characterization of ibrutinib liposils

**2.5.1 Measurement of particle size and zeta potential.** Using the Malvern particle size analyzer (Zetasizer Nano ZSP, model number (ZEN5600)) and the dynamic light scattering technique, the size of nanoliposomes and liposils was determined. Based on the intensity of the light scattering, the software calculated the findings using predetermined methods. The dilution of the dispersions with deionized water was followed by their placement in polystyrene electrophoretic cells for zeta potential measurement. A gold-plated electrode U-shaped cell was used to monitor the zeta potential throughout 100 runs at 25 ˚C at a rate of 250 particles per second. Three times the experiments were conducted.

**2.5.2 Particle morphology.** *2.5.2.1 Inverse phase microscopy.* Nanoliposomes and liposils were observed in phase-contrast mode on an inverted microscope ((Olympus, model -AxioCamERc 5s, Japan). A charge-coupled device (CCD) camera (C3077; Hamamatsu Photonics K. K., Hamamatsu, Japan) was used to record phase-contrast pictures. An image processor was used to evaluate the photos. Each sample underwent a triple analysis.

*2.5.2.2 Transmission electron microscopy.* Transmission electron microscopy (JEM-2000 EXII; JEOL, Tokyo, Japan) has been employed to examine the morphology of the ibrutinib-loaded nanoliposomes and liposils. A film-coated Cu grid was treated with one drop of a 2% (w/v) aqueous phosphotungstic acid solution before being stained with one drop of diluted ibrutinib nanoliposomes. The Cu grid was then let to dry for contrast enhancement. The samples were analyzed using transmission electron microscopy at a 72000x magnification.

**2.5.3 Entrapment efficiency.** The dialysis technique was employed to evaluate the entrapment efficiency of both nanoliposomes and liposils [26]. Each formulation's five milliliters, which included five milligrams of ibrutinib, was put into a dialysis membrane bag with a molecular weight cut-off of 12 to 14 kDa. The sealed bag was immersed in 100 mL of pH 7.4 phosphate buffer containing 1 M sodium salicylate and stirred at 37 ˚C and 100 rpm for one hour. Afterward, the formulation within the dialysis bag was dispersed in a 5 mL mixture of organic solvent and water (50:50 v/v). A 2 mL sample of the media was collected to quantify the amount of free drug present after one hour. A stability-indicating HPLC technique was used to measure the quantity of ibrutinib in the dissolved formulation and the 1-hour sample, and formula 1 was used to compute the percentage of EE.

$$PercentageEntrapmentefficiency = \frac{(Totalamountofibrutinib - Freeibrutinib)}{Totalamountofibrutinib} x100 \qquad (1)$$

**2.5.4 Solid state characterization.** *2.5.4.1 Fourier transform infrared spectroscopy.* Using the KBr disc technique and the Tensor 27 FTIR Spectrophotometer (Bruker Optics, Germany) in the region of 600 to 4000 cm$^{-1}$, with 99.999% nitrogen and a resolution of 4 cm$^{-1}$, ibrutinib, nanoliposomes, and liposils were all submitted to FTIR analysis. The sample pellets were prepared with a compression force of 50 MPa and have the diameter of 10mm.

*2.5.4.2 Powder x-ray diffraction.* Employing Huber Guinier camera G670 (concurrent with Bragg-Brentano geometry), the powder X-ray diffraction patterns of bulk ibrutinib, nanoliposomes, and liposils were captured. A flat aluminium sample holder was utilized to hold the sample, and it was scanned from 5 to 60˚ at a rate of 2˚/min at 2θ intervals.

*2.5.4.3 Differential scanning colorimetry.* Using a Perkin Elmer DSC/7 differential scanning calorimeter (Perkin-Elmer, CT-USA) outfitted with a TAC 7/DX instrument controller, the thermal characteristics of bulk ibrutinib, Nanoliposomes, and liposils were examined. Indium was used to calibrate the instrument's heat of fusionandmelting point. In the temperature range between 30–400 ˚C, a heating rate of 10 ˚C/min was used. An empty pan was utilized as

the reference standard and Standard Perkin-Elmer aluminum sample pans were used. Five-milligram samples were subjected to analyses in triplicate while being $N_2$-purged.

## 2.6 Evaluation of physical stability

**2.6.1 Osmotic stability.** Different osmotic pressures were used to suspend ibrutinib-loaded nanoliposomes and liposil formulations in phosphate buffer solution (pH 7.4). HPLC was used to measure the quantity of drug released at various time intervals.

**2.6.2 Low pH stability.** Ibrutinib loaded nanoliposomes and ibrutinib loaded liposil samples were placed in phosphate buffer (pH 2, 4, 6 & pH 7.4) for 30 minutes. Low pH stability experiments were performed at room temperature. HPLC was used to measure the quantity of drug released at various time intervals.

**2.6.3 Triton X-100 stability.** As described elsewhere, the stability of ibrutinib-loaded liposils and nanoliposomes were estimated in Triton X-100 [27]. The nonionic surfactant Triton X-100 can dissolve biological membranes, interfering with the liposomes' thermodynamic stability. In distilled water, Triton X-100 was produced at various concentrations (0.25%, 0.5%, 1%, 2.5%, 5%, and 10% v/v). A mixture of 9 ml of Triton X-100 solution and 1 ml of the ibrutinib formulation was vigorously shaken at 100 rpm in a water bath maintained at 37 ˚C for one hour. Regarding entrapment efficiency, PDI, particle size, and percent drug release, the stability of ibrutinib nanoliposomes and liposils was assessed.

**2.6.4 Stability study.** Optimized liposil batches were evaluated for stability over 6 months following ICH guidelines at 5 ºC and 25 ºC/60% RH. Samples were freeze-dried with 5% trehalose as a cryoprotectant, stored at -80 ºC, and then freeze-dried at -54 ºC under vacuum. At monthly intervals, samples were reconstituted and assessed for particle size and % EE. The experiment was conducted in triplicate, and results were compared to initial values of the formulations. The solution state stability of ibrutinib formulations was also evaluated at 5 C and 25 ºC in glass vials with respect to particle size and entrapment efficiency. The particle size and entrapment efficiency were evaluated as mentioned above. The experiment was performed in triplicate and the results were compared with initial values of respective formulations.

## 2.7 In vitro cytotoxicity study(3-(4, 5-Dimethylthiazol-2-yl)-2,5-diphenyltetrazolium bromide–MTT Assay)

The cytotoxicity of ibrutinib-loaded nanoliposomes and liposils against the sarcoma (S180) cancer cell line was assessed using the MTT test [23]. The S180 cells were maintained in humidified conditions at a temperature of 37˚C under the influence of CO2 monitored at 5%. The cells were grown in DMEM media which was supplemented with 10% heat inactivated fetal bovine serum (FBS). The cells were harvested at 90% confluency. The ibrutinib formulations were treated with S180 cells for 48 hours at various doses in 96-well plates. Each well was supplied with the MTT solution (5 mg/mL) after 48 h, and each well was then incubated for a further 4 h. After each well had received 150 µL of DMSO, the resulting formazan crystals had been solubilized. A microplate reader (Spectramax 190, Molecular Devices, USA) was used to measure the absorbance of the plates at 570 nm. The percentage cell viability was calculated by comparing the absorbance of the control (untreated) cells to the absorbance of the test cells. The $IC_{50}$ value of the various ibrutinib formulations was obtained by plotting a graph of the log concentration of ibrutinib versus the percentage cell viability.

## 2.8 In-vitrodrug release study

For the *in vitro* release investigation, multi-compartment rotating cells with a dialysis membrane bag (Sartorius cut off 12,000 Da) were utilized. The setup consisted of six compartments.

A dialysis membrane bag was filled with five milliliters of various ibrutinib formulations. The study initiated by placing the end-sealed dialysis bag in 100 mL of pH 7.4 phosphate buffer containing 1 M sodium salicylate, while maintaining a temperature of 37˚C and a stirring speed of 100 rpm. At specified time intervals, two milliliters of the sample were extracted and substituted with an equal volume of fresh-release media. The concentration of ibrutinib in the samples was determined using the HPLC technique. The experiment was conducted in triplicate.

## 2.9 Kinetic analysis

The outcomes of the *In-vitro* release investigation were subjected to fitting with various kinetic models, including Korsemeyer Peppa's, Higuchi's, first-order, and zero-order models. This analysis aimed to elucidate the method and mechanism of drug release. The release of ibrutinib from the nanosponge formulation was determined using a curve fitting approach, where multiple kinetic equations were applied to the data obtained from *in vitro* release assessments.

## 2.10 Bioavailability study protocol

To conduct the oral bioavailability studies of ibrutinib in three formulations namely nanoliposomes, liposils, and free ibrutinib, healthy male Wistar rats (200–250 g) were used. The study was conducted in an animal house under well-ventilated conditions following ethical compliance. The rats were kept in large cages under a 12-h dark and light cycle. The rats were segregated into 4 groups, and each group consisted of 6 animals. The groups were administered with 20 mg/kg of free ibrutinib in suspension form(4mg of ibrutinib in 1ml water), ibrutinib nanoliposomes, or liposil formulations orally. At specific time points the blood samples were collected, ranging from 1 min to 24 h after the administration of the drug. By centrifugation from the blood samples plasma was isolated, and then analyzed for ibrutinib concentration using HPLC. To account for inter-individual variability, a tolbutamide solution was added as an internal standard. All the animal studies were approved by Institute Animal Ethics Committee (IAEC) with Committee for the Purpose of Control and Supervision of Experiments on Animals (CPCSEA) Registration No: 1292/ac09/CPCSEA (Institution Name: Vijaya College of Pharmacy, Hyderabad, India. The IAEC is granted permission by the governmental regulatory organization "Committee for control and supervision of experiments on animals".

**2.10.1 Evaluation of oral bioavailability and pharmacokinetic parameters.** The plasma concentration-time profiles were generated and non-compartmental pharmacokinetic analysis (NCA) was employed to calculate the pharmacokinetic characteristics and oral bioavailability of ibrutinib. The NCA extravascular input mode was used to calculate the PK variables using Microsoft Excel 2010. The linear trapezoidal approach was employed to calculate PK variables like $T_{max}$, $C_{max}$exposure, MRT, $t_{1/2}$, $\lambda z$, Vz, and $C_{l/f}$. after inputting the plasma concentration and time data for each animal along with other information like dosage and units. Additionally, the relative bioavailability ($F_{rel}$) of ibrutinib in liposils and nanoliposomes was calculated by contrasting their exposures found in the PK analyses with those of pure ibrutinib suspension using the following formula.

$$F_{rel} = \frac{AUC(ibrutinib\,laoded\,formulation) * dose\,of\,ibrutinib\,suspension}{AUC(ibrutinib\,suspension) * dose\,of\,ibrutinib\,formulation} \tag{2}$$

**Table 1. Zeta potential, polydispersity index and particle size values.**

| Sample | Zeta potential (mV) | Polydispersity Index | Mean hydrodynamic diameter ± SD (nm) | Entrapment efficiency (%) |
|---|---|---|---|---|
| Nanoliposomes | 18.72 ± 3.18 | 0.17 ± 0.005 | 208.34 ± 5.14 | 89.94 |
| Liposils in suspension | -25.12 ± 1.76 | 0.19 ± 0.005 | 238.36 ± 3.76 | 90.12 |
| Liposils after drying | -29.34 ± 3.14 | 0.22 ± 3.76 | 241.72 ± 8.12 | 89.62 |

(Values were presented as mean±S.D., n = 3 and all calculations were done in triplicate.)

## 2.11 Statistical analysis

The mean ± standard deviation (SD) was used to express the pharmacokinetic parameters. With the aid of the statistical analysis tool GraphPad Prism (GraphPad Software Inc., San Diego, CA), the parameters were further examined. With the aid of GraphPad, the statistical significance of the differences between the means was examined using the unpaired, two-sample student t-test. Mean differences were considered significant or highly significant when the corresponding P values were less than 0.05 and 0.01, respectively.

## 3. Results

### 3.1 Characterization of ibrutinib liposils

**3.1.1 Measurement of particle size and zeta potential.** Ibrutinib liposils have a low polydispersity index of 0.22 and a narrow size distribution of about 240 nm, as seen in Table 1. Freshly made nanoliposomes exhibited a polydispersity index of 0.17 and a diameter of 208 nm. The size distribution of the liposil spheres that were still in suspension was centered at 238 nm, and their polydispersity index was just 0.19. Analysis reveals varying surface charge characteristics, with nanoliposomes displaying a slight positive zeta potential of 18.72 ± 3.18 mV, while liposils in suspension and after drying exhibit negative zeta potentials of -25.12 ± 1.76 mV and -29.34 ± 3.14 mV, respectively. This suggests differing electrostatic interactions, with implications for stability and cellular interactions.

**3.1.2 Particle morphology.** The morphology of both nanoliposomes and liposils was examined using inverse phase microscopy and transmission electron microscopy. The micrographs obtained through inverse phase microscopy revealed that the particles were spherical in shapes, as illustrated in Fig 1A & 1B. The nanoliposomes were observed to have a diameter in the range of 200–210 nm, while the liposils were found to be larger in size, ranging from 230–250 nm. These results were congruent with the findings those derived from dynamic light scattering. TEM images of the liposils (Fig 1C & 1D) showed that the nanocapsules had a mean diameter similar to that of unilamellar nanoliposomes (~ 250 nm). Additionally, the silica-coated liposomes appeared more spherical than the conventional nanoliposomes, indicating that the silica shell may have provided external structural reinforcement to the ultra-deformable lipid bilayer.

**3.1.3 Entrapment efficiency.** The comparative scrutiny of the entrapped drug quantities within liposils and nanoliposomes, elucidated through the data provided in Table 1, underscores compelling parallels. Suspensions of both liposils and nanoliposomes showcase robust entrapment efficiencies, hovering around an impressive 90%. Particularly noteworthy is the post-drying observation, wherein liposils maintain commendable entrapment efficiency, registering at 89.62%. This sustained and elevated level of entrapment efficacy accentuates the proficiency of drug encapsulation within these sophisticated delivery systems, thereby underscoring their potential for efficacious therapeutic interventions.

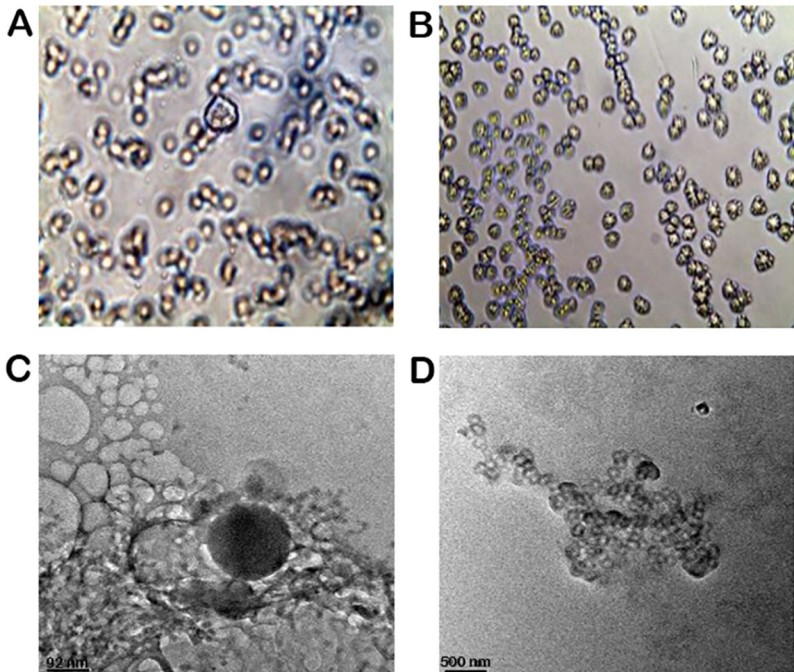

**Fig 1.** Inverse phase micrographs **(A)** Ibrutinib nanoliposomes and **(B)** Ibrutinib liposils. TEM images **(C)** Ibrutinib nanoliposomes and **(D)** Ibrutinib liposils.

**3.1.4 Solid state characterization.** *3.1.4.1 Fourier transform infrared spectroscopy*. FTIR spectroscopy was conducted to verify the presence of silica coating on the nanoliposomes. The FTIR spectra of Ibrutinib, Nanoliposomes, and Liposil formulation are displayed in Fig 2A. Several distinctive peaks could be seen in the FTIR spectra of plain Ibrutinib, including C-N vibrations at 1244 cm$^{-1}$, aromatic C=N and C=C stretching vibrations between 1500 and 1587 cm$^{-1}$, aromatic C-H stretching vibrations at 3065 cm$^{-1}$, and N-H stretching vibrations between 3600 and 3200 cm$^{-1}$[25]. The FTIR spectra of nanoliposomes revealed two distinctive peaks at 1087 cm$^{-1}$ and 2924 cm$^{-1}$, respectively, owing to the P-O of phosphotidylcholine and methylene C-H antisymmetric stretching. The creation of the silica shell was verified by the presence of vibrational absorption bands in the liposil FTIR spectra around 470 cm$^{-1}$ (Si-O bending), 859 cm$^{-1}$ (Si-O-Si bending vibration),and 1080 cm$^{-1}$ (asymmetric Si-O-Si stretching). The characteristic peaks of ibrutinib in both formulations were shifted, indicating the encapsulation of ibrutinib in liposomes. The characteristic peaks of the drug are in concurrence with the reported values [21, 28, 29].

*3.1.4.2 Powder x-ray diffraction*. The XRPD results (Fig 2B) of ibrutinib showed sharp, characteristic crystalline peaks at 2θ of 28.9˚, 21.3˚, 19.0˚, 16.1˚, 13.7˚, 12.3˚, 10.6˚, 5.7˚, etc., representing the plain ibrutinib crystalline nature. In both formulations, disappearance of the characteristic sharp crystalline peaks of the drug, implying that ibrutinib was entirely and effectively encapsulated into the core of the liposome bilayer structure. This outcome aligns with the FTIR results, further validating the drug's internalization.

*3.1.4.3 Differential scanning colorimetry*. The melting point of ibrutinib, 159.12 ˚C, is where the drug shows an intense endothermic peak [26] (Fig 2C). The disappearance of the sharp characteristic peak of ibrutinib in both the formulations implies the ibrutinib amorphous encapsulated dispersion.

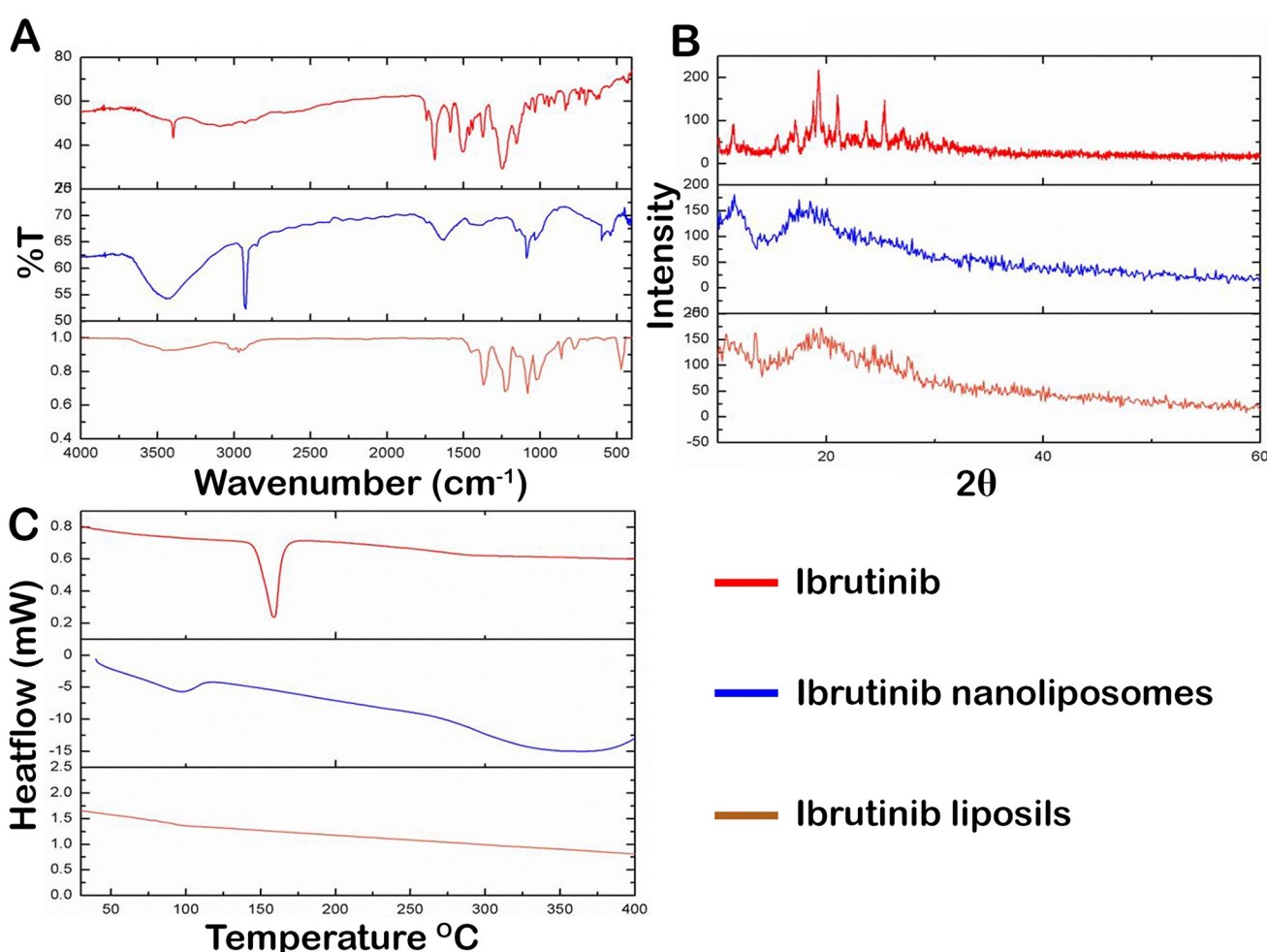

**Fig 2. Solid state characterization of ibrutinib, nanoliposomes and liposils. (A)**FTIR spectra of plain ibrutinib, nanoliposomes and liposils formulation. **(B)** XRPD pattern of plain ibrutinib, nanoliposomes and liposils formulation. **(C)** DSC thermograms of plain ibrutinib, nanoliposomes and liposils formulation.

## 3.2 Evaluation of physical stability

**3.2.1 Osmotic stability.** In order to detect drug release from the internal structure using HPLC, the physical stability of liposils and nanoliposomes loaded with ibrutinib was assessed in phosphate buffer (pH 7.4) under various osmotic pressures. The response of liposils to different osmotic pressure buffer media was studied and compared to that of nanoliposomes under the same conditions (Fig 3A). The results demonstrate that there is negligible drug release from liposils to osmotic pressure, in contrast to the significant drug release observed for nanoliposomes. These findings confirm that the silica coating stabilizes the liposomes and controls drug release during changes in osmolarity, which is a critical requirement for oral drug delivery applications. This suggests that the structures will withstand the different digestive system organs.

**3.2.2 Low pH stability.** The study systematically investigated and contrasted the stabilities of ibrutinib-loaded nanoliposomes and liposils across a spectrum of pH conditions (2, 4, 6, and 7.4) over a defined period of 30 minutes at ambient room temperature. Notably, the data

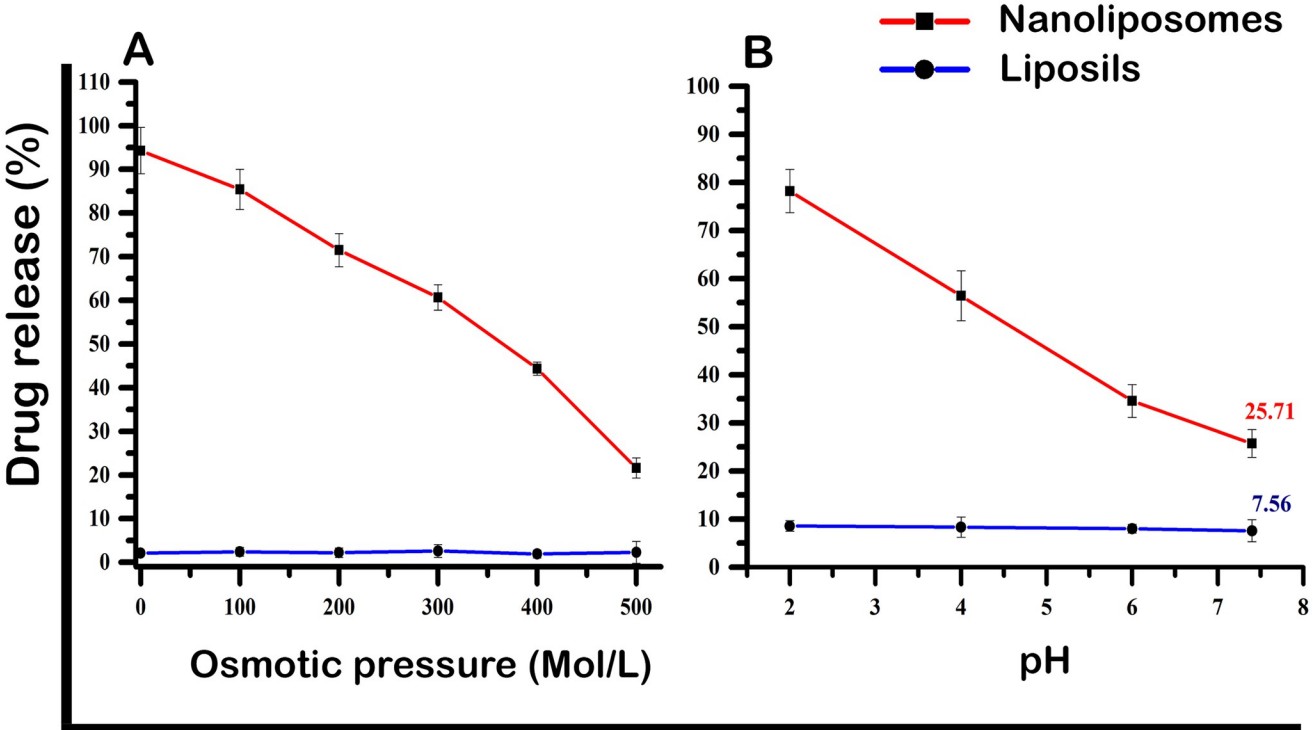

**Fig 3.** **(A)** Impact of osmotic pressure on drug release from nanoliposomes and liposils formulation. **(B)** Effect of external pH on the release of drug incorporated into nanoliposomes and liposils formulation.

unveiled the susceptibility of nanoliposomes to pH-induced perturbations, contrasting sharply with the robust resilience exhibited by liposils (Fig 3B).

**3.2.3 Triton X-100 stability.** To check the physical stability of ibrutinib loaded nanoliposomes and liposils, both the formulations were exposed to different concentrations of solubilizing agent (0.1%, 0.25%, 0.5%, 1%, 2.5%, 5% and 10% v/v). The stability of prepared ibrutinib formulations were assessed with respect to entrapment efficiency and particle size of liposils. Fig 4A–4C shows the variation in particle sizes, polydispersity index and entrapment efficiencies respectively. It was shown that the addition of Triton X-100 significantly reduced nanoliposome stability. The nanoliposome particle size somewhat changed by changing Triton X-100 concentrations from 0.1 to 0.5%, while the entrapment efficiency and polydispersity index mainly remained unaltered. This might be because surfactant molecules are incorporated into and around the liposomes' bilayer, increasing the liposomes size. When the concentration of Triton X-100 reached 1%, there was a significant decrease in both the entrapment efficiency and particle size, accompanied by a sharp increase in the PDI. These changes suggest the formation of micelles and the initiation of phospholipid bilayer disruption in the liposomes. Consequently, the particle size and entrapment efficiency decreased, leading to the formation of a heterogeneous dispersion. Increased Triton X-100 concentration reduced particle size to just 102 nm and a high PDI, which showed that the liposomes had been completely destroyed and the compounds they contained had been micellized.

Silica coated liposomes did not experience any notable changes in entrapment efficiency, particle size and PDI until Triton X-100 concentration reached 5% v/v, in contrast to ibrutinib nanoliposomes. At Triton X-100 concentrations over 5% v/v, there was a modest rise in the

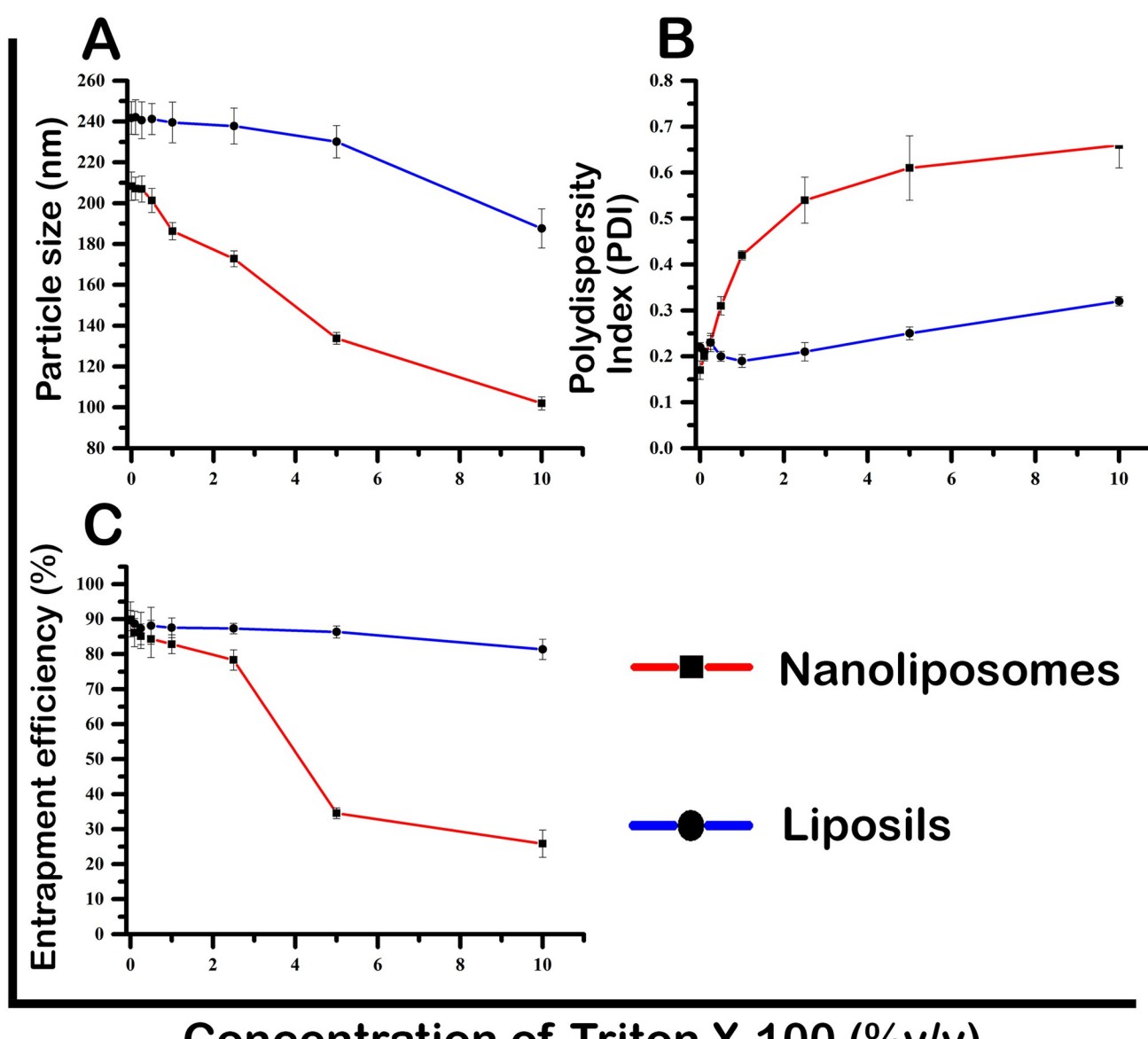

**Fig 4. (A)** Variation of particle size of both the formulations at various concentrations of Triton X-100.**(B)**Variation of polydispersity index (PDI) of both the formulations at various concentrations of Triton X-100.**(C)**Variation of entrapment efficiency of both the formulations at different concentrations of Triton X-100.

PDI and a minor decrease in the entrapment efficiency and particle size of liposils. The data clearly show that ibrutinib liposils outperform uncoated liposomes in withstanding Triton X-100's disruptive action. This greater stability over traditional liposomal systems is due to the silica coating.

**3.2.4 Stability study.** The stability of lyophilized formulations of ibrutinib nanoliposomes and liposils was determined by computing entrapment efficiency and particle size after storage at 25 ℃and 5 ℃/60% RH. According to the findings, neither formulation's particle size nor entrapment effectiveness significantly changed after six months at 5 ℃. However, at 25 ℃/ 60% RH, the entrapment efficiency of ibrutinib nanoliposomes decreased after 3 months,

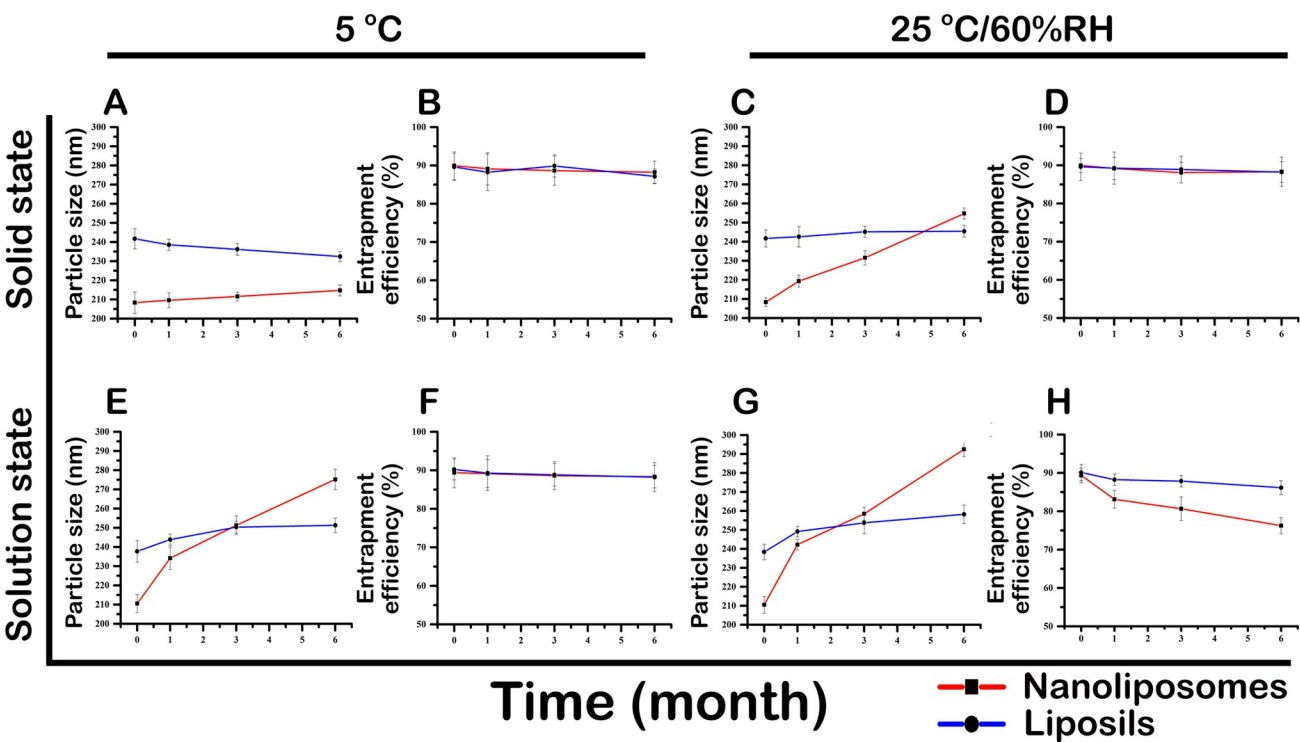

**Fig 5.** Results of Solid-state stability studies of **(A)** Lyophilized ibrutinib nanoliposomes and **(B)** Liposils stored at 5 °C**(C)** Lyophilized ibrutinib nanoliposomes and **(D)** Liposils stored at 25 °C/60% RH with respect to particle size and entrapment efficiency. Results ofSolution state stability studies of **(E)** Lyophilized ibrutinib nanoliposomes and **(F)** Liposils stored at 5 °C**(G)** Lyophilized ibrutinib nanoliposomes and **(H)** Liposils stored at 25 °C/60% RH with respect to particle size and entrapment efficiency.

indicating drug leakage and increased particle size due to particle aggregation. Conversely, particle size and entrapment efficiency remained stable for the entire 6-month duration at 25 °C/60% RH. This is illustrated in Fig 5A–5D.

The solution state stability was evaluated at 5 °C and at 25 °C/60% RH, to better understand the stability of ibrutinib formulations. Under refrigerated conditions the liposil solution was found to be stable for three months, while the nanoliposomes formulation was stable for only 15 days. At 25 °C/60% RH, both formulations showed significant entrapment efficiency and particle size changes. However the liposil formulation did not exhibit any significant changes in entrapment efficiency and size for up to 1 month, the liposomal solution precipitated and exhibited an increase in particle size and entrapment efficiency after 7 days. These results confirm that liposils are more stable for longer periods than conventional liposomes due to a rigid silica layer (Fig 5E–5H). These findings are in concurrence with the findings of physical stability assessment.

### 3.3 In vitro cytotoxicity study

To assess the cytotoxicity of ibrutinib formulations on the sarcoma (S180) cancer cell line, the MTT assay was used. The percent cell viability was estimated and the inhibitory ratios were found to be dose dependent for all formulations. Fig 6A shows the cell viability curves of S180 cells exposed to free ibrutinib, ibrutinib liposils and nanoliposomes and their corresponding blanks. The $IC_{50}$ values were found to be least for ibrutinib (23.45 ±1.37 μM), while free ibrutinib and ibrutinib liposils had higher $IC_{50}$ values of 44.56 ± 2.43 μM and 53.19 ±3.12 μM, respectively.

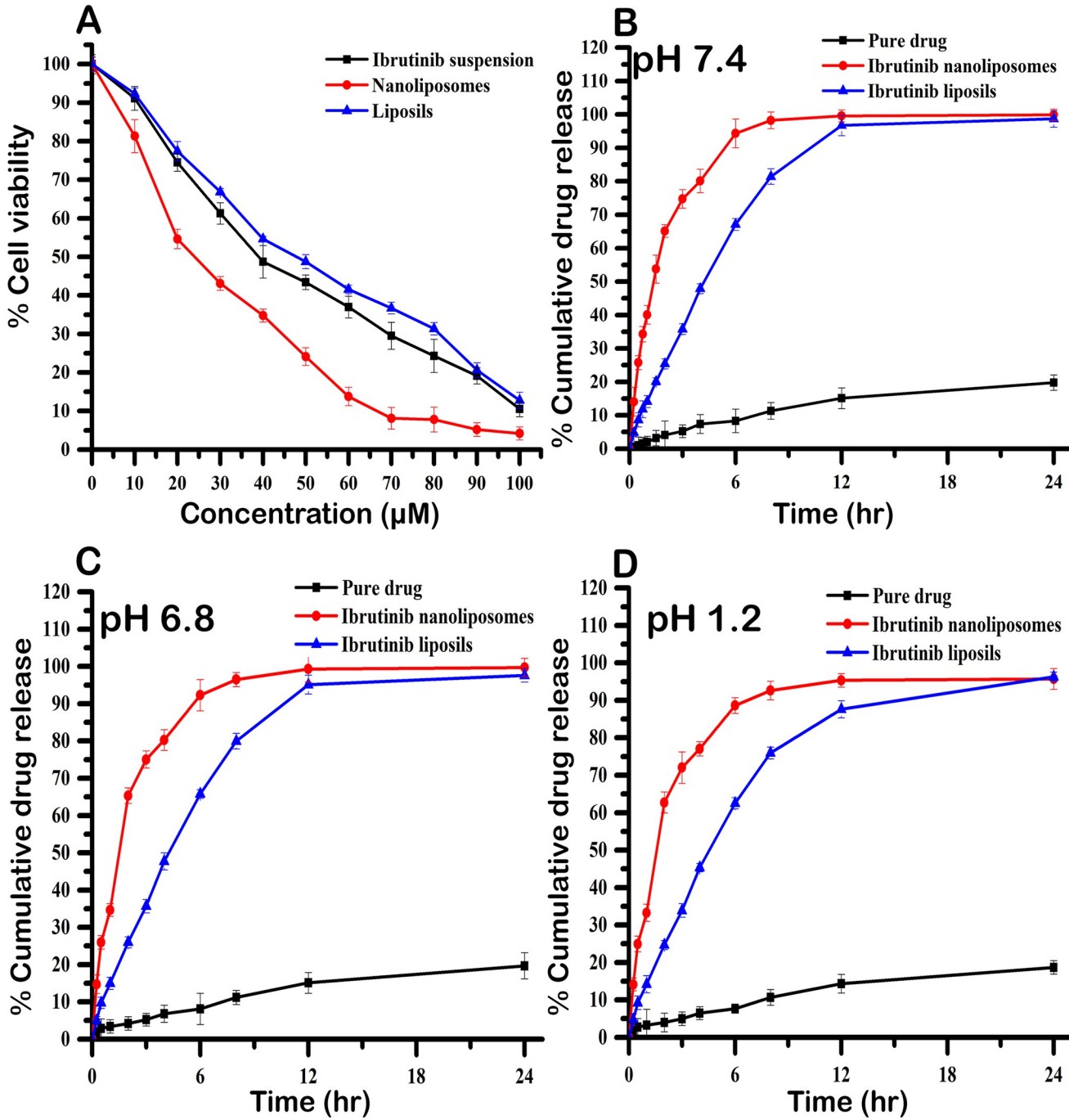

**Fig 6. (A)** Cell viability of S180 tumor cells at various formulation concentrations and **(B)** *in vitro* release profile of ibrutinib from different formulations at pH 7.4 over a period of 48 hours. **(C)** *in vitro* release profile of ibrutinib from different formulations at pH 6.8 over a period of 48 hours. **(D)** *in vitro* release profile of ibrutinib from different formulations at pH 1.2 over a period of 48 hours.

### 3.4 In-vitro drug release study

In a phosphate buffer (pH 7.4) with 1 M sodium salicylate, the release profile of several ibrutinib formulations was assessed. The graph in Fig 6B indicates the cumulative % of drug released over time for free ibrutinib, ibrutinib nanoliposomes, and liposils. Initially, all formulations

exhibited similar release patterns, potentially due to the presence of free drug (~5%). After 24 hours, free ibrutinib only released 8% of the drug. In contrast, ibrutinib loaded in liposomes had a delayed release, with 82% of the drug released within 6 hours. The liposils' silica coating progressively destroys over time, releasing the drug that was entrapped for longer. This was demonstrated by the ibrutinib liposil concentration-time profile, which revealed that the whole drug release occurred within 48 hours. Thus, by altering the sol-gel process reaction conditions, the silica coating reduced the release rate and made it easier to regulate or customize the release of the drug from the liposomes. Furthermore, to comprehensively delineate the nuanced efficacy disparities, supplementary pH stability analyses were conducted at pH 6.8 and pH 1.2, underscoring the discernibly superior stability profile of ibrutinib liposils, a conclusion substantiated by the meticulously quantified drug release kinetics meticulously catalogued in Fig 6C & 6D. These findings underscore the pivotal role of liposils as promising candidates in drug delivery systems, particularly in navigating diverse pH environments with enhanced stability and efficacy.

### 3.5 Kinetic analysis

Different mathematical models were used to assess the mechanism of drug release from the formulation of liposils. In order to ascertain the order and mechanism of drug release, the drug release data for the liposils formulation was fitted into several kinetic models (Fig 7A–7D). The release exponent, denoted as "n," in the Korsemeyer Peppas model was determined to be 0.231, which indicated aberrant behaviour as it deviated from the expected range of $0.43 < n$. The correlation coefficient ($R2 = 0.7224$) suggested a diffusion-based release mechanism (Table 2).

### 3.6 Bioavailability study protocol

Healthy male Wistar rats were used to conduct oral bioavailability studies of ibrutinib nanoliposomes, liposils, and free ibrutinib. Ibrutinib was administered in a dosage of 20 mg/kg, and the mean and standard error of the mean were statistically computed for various pharmacokinetic parameters. Fig 8 displays the corresponding plasma concentration-time curves of ibrutinib nanoliposomes, liposils, and free ibrutinib suspension after oral administration. The corresponding pharmacokinetic parameters are presented in Table 3. Ibrutinib levels in plasma were noticeable for about 48 hours for ibrutinib liposils, 24 hours for ibrutinib nanoliposomes, and 6 hours for ibrutinib suspension. The maximum absorption ($C_{max}$) values for ibrutinib suspension, ibrutinib nanoliposomes, and ibrutinib liposils were found to be $122.52 \pm 4.56$ ng/mL, $436.74 \pm 19.96$ ng/mL, and $352.43 \pm 15.54$ ng/mL, respectively. The $C_{max}$ values of both liposomal formulations were about three times higher than that of ibrutinib suspension, and the difference was highly significant ($P < 0.01$). Precipitation or poor solubility at the absorption site might cause the suspension's reduced absorption. The time taken for $C_{max}$ ($1.56 \pm 0.12$ h, $2.12 \pm 0.23$ h, and $2.92 \pm 0.54$ h for ibrutinib suspension, nanoliposomes, and liposils, respectively), absorption from ibrutinib suspension was quicker than that of both liposomal formulations. This difference in absorption time was noteworthy ($P < 0.05$). The half-life values were 4.82 hours, 8.35 hours, and 12.67 hours for ibrutinib suspension, nanoliposomes, and liposils. The $t_{1/2}$ of liposils was high because of the sustained release of drug provided by the silica coating over a longer duration of time. Other pharmacokinetic parameters are presented in Table 4. Overall, the oral bioavailability of ibrutinib liposils and nanoliposomes increased by 4.08 and 3.12 folds, respectively, compared to ibrutinib suspension. When a drug is given as liposils, these pharmacokinetic factors can assist maintain the plasma concentration longer and increase its effectiveness.

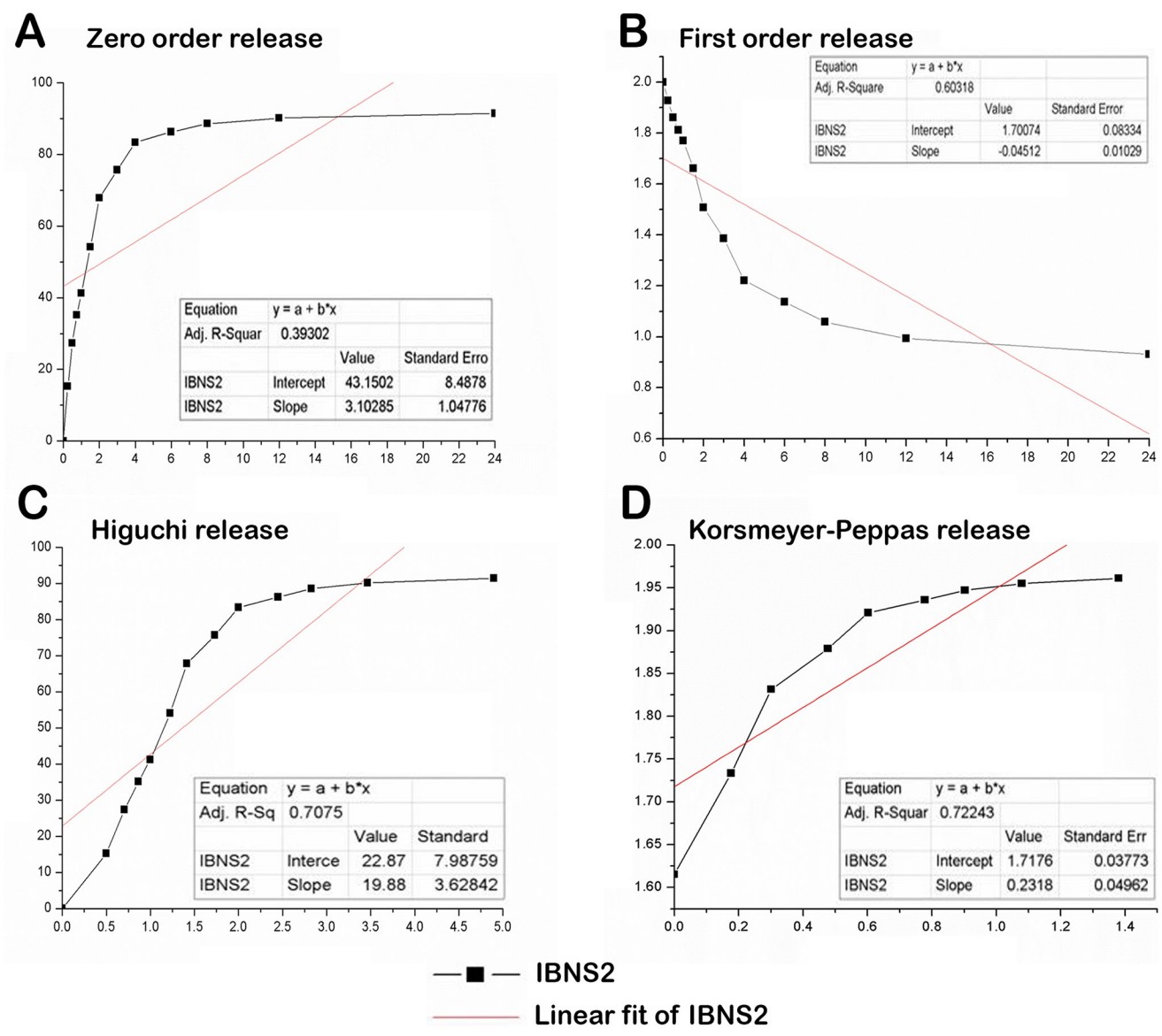

**Fig 7.** Kinetic studies for **(A)** Zero order release, **(B)** First order release, **(C)** Higuchi release and **(D)** Korsmeyer-Peppas of liposils formulations.

## 4. Discussion

In the realm of pharmaceutical science, safeguarding drug efficacy and protecting against gastric degradation are paramount concerns, particularly for soft drug delivery systems. Among the diverse strategies emerging in recent times, silica-coated drug delivery techniques have garnered considerable attention for their potential to preserve drug activity and shield against the

**Table 2. Release kinetics of ibrutinib liposils formulation.**

| Korsmeyer-Peppas | | Higuchi | | First Order | | Zero Order | |
|---|---|---|---|---|---|---|---|
| R² | n | R² | n | R² | n | R² | n |
| 0.7224 | 0.231 | 0.7075 | 19.88 | 0.6031 | -0.045 | 0.3930 | 3.102 |

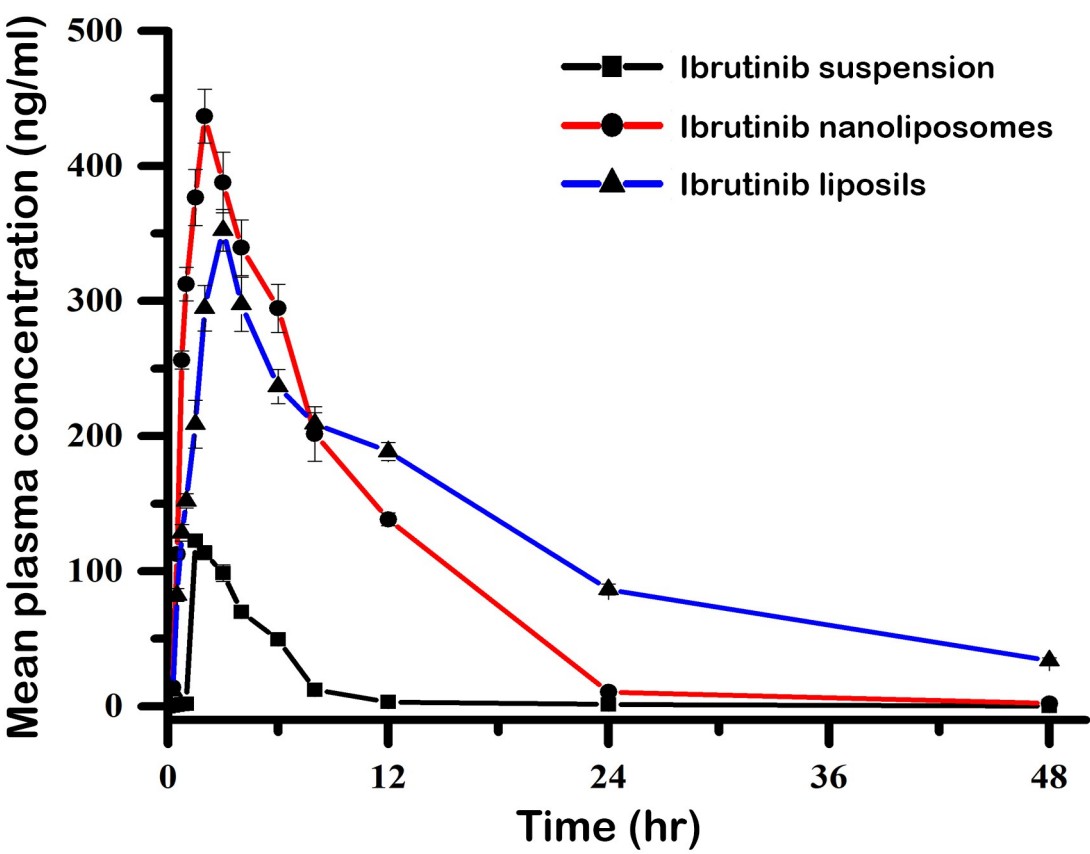

**Fig 8. Mean Plasma drug concentration versus time profile curves of free ibrutinib, ibrutinib nanoliposomes and ibrutinib liposils after oral administration in rats.**

**Table 3. Plasma concentration time profile of ibrutinib.**

| Time in hr | Mean plasma concentration (ng/ml) | | |
|---|---|---|---|
| | Ibrutinib suspension (Mean ± SE) | Ibrutinib nanoliposomes (Mean ± SE) | Ibrutinib liposils (Mean ± SE) |
| 0 | 0 | 0 | 0 |
| 0.25 | 0.514 ± 0.18 | 14.112 ± 0.83 | 11.548 ± 1.16 |
| 0.5 | 1.013 ± 0.42 | 112.765 ± 3.46 | 82.54 ± 4.78 |
| 0.75 | 1.611 ± 0.28 | 256.341 ± 6.69 | 128.46 ± 6.17 |
| 1 | 2.013 ± 0.22 | 312.543 ± 12.35 | 152.134 ± 5.36 |
| 1.5 | 122.52 ± 4.56 | 376.546 ± 20.72 | 208.928 ± 17.72 |
| 2 | 113.78 ± 5.12 | 436.742 ± 19.96 | 294.76 ± 16.89 |
| 3 | 98.56 ± 6.12 | 387.82 ± 22.43 | 352.43 ± 15.54 |
| 4 | 69.87 ± 3.72 | 339.45 ± 20.56 | 297.65 ± 20.14 |
| 6 | 49.46 ± 1.19 | 294.56 ± 17.84 | 236.78 ± 12.69 |
| 8 | 12.34 ± 2.12 | 201.64 ± 20.28 | 209.16 ± 8.28 |
| 12 | 3.22 ± 0.68 | 138.42 ± 4.78 | 188.72 ± 6.74 |
| 24 | 1.42 ± 0.54 | 10.56 ± 2.29 | 86.53 ± 3.99 |
| 48 | 0.23 ± 0.12 | 2.12 ± 0.44 | 33.87 ± 2.12 |

Values were presented as mean ± S.D., n = 3 and all calculations were done in triplicate.

**Table 4. Mean pharmacokinetic parameters of ibrutinib after oral administration of ibrutinib suspension, ibrutinib nanoliposomes and ibrutinib liposils.**

| Pharmacokinetic parameter | Ibrutinib nanoliposomes (Mean ± SE) | Ibrutinib suspension (Mean ± SE) | Ibrutinib liposils (Mean ± SE) |
|---|---|---|---|
| MRT (h) | 14.46 ± 3.78 | 10.32 ± 88.92 | 17.18 ± 4.36 |
| $t_{1/2}$(h) | 8.35 ± 2.94 | 4.82 ± 1.32 | 12.67 ± 3.12 |
| $\Lambda z$ ($h^{-1}$) | 0.13 ± 0.07 | 0.21 ± 0.09 | 0.08 ± 0.054 |
| $AUC_{0 \to \infty}$ (ng.h/ml) | 3456.66 ± 312.18 | 1056.73 ± 88.92 | 4243.91± 478.84 |
| $AUC_{0 \to 48}$ (ng.h/ml) | 3167.82 ± 438.76 | 986.48 ± 102.12 | 4086.52± 318.92 |
| $T_{max}$ | 2.12 ± 0.23 | 1.56 ± 0.12 | 2.92 ± 054 |
| $C_{max}$ (ng/ml) | 436.742 ± 19.96 | 122.52 ± 4.56 | 352.43 ± 15.54 |
| Relative bioavailability (%) | 312.56 | 100 | 408.39 |

Values were presented as mean ± S.D., n = 3 and all calculations were done in triplicate.

influences of gastric enzymes and pH variations [30, 31]. These silica-based drug carriers offer a spectrum of capabilities, including targeted and controlled drug release, contributing to their increasing prominence [32, 33]. A notable innovation in this field is the development of "Liposil," a hybrid approach that merges liposomes with silica, capitalizing on the advantageous attributes of both components [34, 35]. Liposil presents a unique combination of liposomes' exceptional stability and the adaptability of drug loading associated with silica, positioning it as a promising contender among novel drug delivery carriers. The stability of Liposil is facilitated by interfacial hydrogen bonding between the phospholipid head groups and the silica surface, ensuring the integrity and longevity of the hybrid structure, thus enhancing its suitability for innovative drug delivery applications.

Nanoliposomes containing ibrutinib were crafted through reverse phase evaporation, employing carefully optimized parameters. These nanoliposomes, utilized as the foundation for subsequent silica coating, featured choline head groups and zwitterionic phospholipids. The creation of liposils entailed the condensation and hydrolysis of tetraethoxy silane at the surface of unilamellar liposomes, culminating in the formation of robust silica spheres, each boasting an approximate diameter of 240 nm. The underlying mechanism governing nanoliposome templating was the strong electrostatic interaction between negatively charged silica monomers and the positively charged lipid head groups (quaternary ammonium). This interaction served as the driving force behind the templating process. The resultant silica coating displayed exceptional solidity and impermeability. To generate liposils, stable liposomes were employed as templates, while tetraethoxy silane functioned as the source of silicon. This orchestrated a precise synthesis of silica nanoshells on the surface of the template liposomes, a process achieved through spatially controlled sol-gel chemistry. Throughout each stage of the liposil preparation process, dynamic light scattering measurements were conducted, revealing a notable absence of particle aggregation as long as the liposils remained in suspension. The maintenance of this uniform size distribution proves crucial in achieving monodispersed liposil samples. Importantly, the drying step exhibited no adverse impact on the spherical shape and structural integrity of individual liposils. The process of coating nanoliposomes with silica yielded a distinctive core-shell structure, characterized by a negative zeta potential attributable to the presence of net negative charges from free silanol groups. Furthermore, the polydispersity index of the liposil system remained consistent both before and after silica encapsulation, serving as evidence of a homogeneously applied coating, free from any residual silica precipitation or crystallization. Further, the incorporation of a silica coating had no impact on the entrapment efficiency, as it left the internal structure of the phospholipid bilayer, where the

drug resides within hydrophobic pockets, unaffected. Moreover, the silica coating was expected to act as a barrier, preventing drug leakage from the liposomal structure and thereby ensuring the prolonged stability of the liposomes.

The evaluation of the biophysical stability of liposomes can be effectively conducted through the dispersion of phospholipid vesicles in the presence of a surfactant. Triton X-100, a non-ionic surfactant, is constituted by an aromatic hydrocarbon lipophilic moiety and a hydrophilic polyethylene oxide chain. This surfactant finds extensive utility in disrupting cell membranes, especially those comprised of phospholipid bilayer, along with various cellular structures. The propensity of liposomal formulations to aggregate in hydrophobic environments is intricately linked to their concentration levels. However, the introduction of a silica coating surrounding the liposomes is anticipated to mitigate this phenomenon by establishing a robust framework that encapsulates the lipid vesicles, thereby enhancing their stability in such environments.

The discernible contrast in cytotoxicity between free ibrutinib and ibrutinib liposils in comparison to conventional nanoliposomes can be attributed to the incorporation of cationic surface charges facilitated by stearylamine within the bilayer membrane. This characteristic, known for its ability to augment cytotoxic effects across diverse cell lines, presents a significant factor. Unlike conventional nanoliposomes, where the absence of coating exposes the cationic surfaces of ibrutinib liposomes directly to cells, the presence of a protective silica coating in ibrutinib liposils effectively mitigates the impact of these cationic charges. During the in-vitro drug release investigations performed in a phosphate buffer solution maintained at pH 7.4 and supplemented with 1 M sodium salicylate, the initial release kinetics showed a comparable trend among the formulations under study. However, after a 24-hour timeframe, free ibrutinib demonstrated a release rate of merely 8%, contrasting sharply with the considerably higher release rate of 82% exhibited by ibrutinib encapsulated within liposomes, within a mere 6-hour duration. Remarkably, liposils, characterized by their silica coating, exhibited a gradual release profile over duration of 48 hours. This illustrates the potential for regulating drug release from liposomes through the modulation of the sol-gel process and manipulation of the silica coating, offering promising prospects for tailored drug delivery systems.

The analysis of kinetic models reveals distinct patterns in drug release. Notably, the Korsmeyer-Peppas model displays a superior regression coefficient value, indicative of exponential drug release over time. Conversely, both the zero-order and first-order equations exhibit reduced linearity when plotted. Utilizing various mathematical models, such as Korsmeyer-Peppas and Higuchi plots, facilitates a deeper comprehension of the drug release process. In particular, the "n" value within the Korsmeyer-Peppas model characterizes the drug release mechanism. When "n" falls below 0.45 ($0.45 \leq n$), it signifies Fickian diffusion. If "n" falls within the range of $0.45 < n < 0.89$, it points to non-Fickian transport, often described as relaxational transport. Finally, if "n" surpasses 0.89, it suggests super case II transport. In the current scenario, the obtained "n" value of 0.231 from the Korsmeyer-Peppas plots indicates a Fickian diffusion mechanism, implying that the drug release involves both diffusion and erosion processes.

Oral bioavailability studies conducted on healthy male Wistar rats revealed significant differences in pharmacokinetic parameters among ibrutinib nanoliposomes, liposils, and free ibrutinib suspension. Through comprehensive analysis of plasma concentration-time curves, it became evident that the liposomal formulations, especially liposils, exhibited a prolonged and sustained presence of ibrutinib in circulation compared to the conventional suspension. Noteworthy was the discernible extension in drug concentration duration, lasting up to an impressive 48 hours for liposils, 24 hours for nanoliposomes, and merely 6 hours for the suspension. Furthermore, the liposomal formulations demonstrated substantially elevated

maximum absorption ($C_{max}$) values, indicative of a pronounced enhancement in bioavailability relative to the suspension. The observed delayed absorption kinetics and the ensuing protracted half-life with liposils strongly suggested a sustained release mechanism facilitated by the unique silica coating. These compelling findings not only underscored the considerable promise of liposomal formulations, particularly liposils, in augmenting the oral bioavailability of ibrutinib but also hinted at their potential to prolong therapeutic efficacy, thereby offering substantial advantages in the realm of oncological therapeutics. Such insights hold profound implications for clinical translation, urging further exploration and optimization of these innovative drug delivery systems for enhanced patient outcomes.

## 5. Conclusion

Liposomes, owing to the dynamic nature of their phospholipid membranes, face notable challenges in accommodating bulky hydrophobic molecules like Ibrutinib. Despite their myriad advantages, including commendable attributes such as in vivo performance, scalability, biocompatibility, and versatility, liposomes often exhibit a degree of instability that trails behind certain other nanoscale formulations. This study underscores the paramount importance of silica coating on liposils (templated liposomes) in fortifying their physical stability and resilience against disruptive agents such as Triton X-100, as convincingly demonstrated by meticulously conducted stability assessments. The results of comprehensive pharmacokinetic evaluations in rats unequivocally attest to a remarkable enhancement in drug bioavailability upon administration via Liposils. These findings not only underscore the significant promise of Liposils as versatile drug carriers, capable of accommodating a broad spectrum of compounds irrespective of their hydrophilicity or hydrophobicity, but also highlight their independence from the intrinsic stability characteristics of the loaded drug molecules. Moreover, the protective mechanism offered by Liposils, characterized by attributes such as non-toxicity, biodegradability, and compatibility with oral administration, further solidifies their standing as an exceptionally attractive option for a myriad of drug delivery applications in both biomedical research and clinical practice alike.

## Author Contributions

**Conceptualization:** S. Selvamuthukumar.

**Data curation:** Fareeaa Ashar.

**Funding acquisition:** Asif Ansari Shaik Mohammed.

**Methodology:** Fareeaa Ashar.

**Project administration:** S. Selvamuthukumar.

**Resources:** S. Selvamuthukumar.

**Software:** Fareeaa Ashar.

**Supervision:** S. Selvamuthukumar.

**Validation:** Fareeaa Ashar.

**Visualization:** Fareeaa Ashar.

**Writing – original draft:** Fareeaa Ashar.

**Writing – review & editing:** Fareeaa Ashar, Asif Ansari Shaik Mohammed, S. Selvamuthukumar.

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
