## [Decision Letter · Decision Letter 0]

10 Mar 2024

PONE-D-24-01636Enhancement of Oral Bioavailability of Ibrutinib Using Liposil Nanohybrid Delivery SystemPLOS ONE

Dear Dr. Selvamuthukumar,

Thank you for submitting your manuscript to PLOS ONE. After careful consideration, we feel that it has merit but does not fully meet PLOS ONE’s publication criteria as it currently stands. Therefore, we invite you to submit a revised version of the manuscript that addresses the points raised during the review process.

We look forward to receiving your revised manuscript.

Kind regards,

Pradeep Kumar, Ph.D.

Academic Editor

PLOS ONE

2. PLOS requires an ORCID iD for the corresponding author in Editorial Manager on papers submitted after December 6th, 2016. Please ensure that you have an ORCID iD and that it is validated in Editorial Manager. To do this, go to ‘Update my Information’ (in the upper left-hand corner of the main menu), and click on the Fetch/Validate link next to the ORCID field. This will take you to the ORCID site and allow you to create a new iD or authenticate a pre-existing iD in Editorial Manager. Please see the following video for instructions on linking an ORCID iD to your Editorial Manager account: https://www.youtube.com/watch?v=_xcclfuvtxQ.

Reviewers' comments:

Reviewer's Responses to Questions

**Comments to the Author**

1. Is the manuscript technically sound, and do the data support the conclusions?

Reviewer #1: Yes

Reviewer #2: No

2. Has the statistical analysis been performed appropriately and rigorously? 

Reviewer #1: Yes

Reviewer #2: No

3. Have the authors made all data underlying the findings in their manuscript fully available?

Reviewer #1: Yes

Reviewer #2: Yes

4. Is the manuscript presented in an intelligible fashion and written in standard English?

Reviewer #1: No

Reviewer #2: No

5. Review Comments to the Author

Reviewer #1: The manuscript entitled “Enhancement of Oral Bioavailability of Ibrutinib Using Liposil Nanohybrid Delivery System” deals with the formulation and in vitro as well as in vivo characterization of silica-coated nanoliposomes loaded with ibrutinib. The manuscript contains interesting and novel information about the liposome development, however I have few concerns, which should be addressed before further decision.

1.Please check the grammar throughout the manuscript.

2.Do the authors expect silicosis due to the silica coating?

3.Please specify the type of DLS equipment applied for the measurements!

4.Please specify the parameters of the disc prepared for FTIR! (diameter, applied compression force etc.)!

5.What was the reason for using 1M sodium salicylate in the release medium? Please explain!

6.The nanoliposomes showed positive zeta potential. Can this have an effect on the absorption?

7.Besides the drug encapsulation efficacy (EE) the drug loading (DL) is also an important parameter. Please provide it!

8.Based on the DSC and XRPD results the drug went through amorphization. How can we be sure, that the drug is inside the nanoparticle?

9.In case of Fig. 2A what does CM mean on the x-scale? Usually, wavenumber is plotted on the x-scale of FTIR curve.

10.Standard deviations are missing from the Figures! I guess there were more parallel experiments performed.

11.What is the target concentration of ibrutinib? In the MTT studies it is visible, that above 10 µM the formulations show cytotoxicity.

Reviewer #2: The submitted manuscript provides for the development and analysis of silica-coated Ibrutinib-loaded nanoliposomes (Liposils) to enhance the bioavailability of Ibrutinib. Numerous in vitro characterization studies have been undertaken prior to in vivo analysis of the developed system in a Wistar rat model. The results show that the platform increases the bioavailability of the loaded drug with the liposils noted to be stable at ICH conditions. There are however a few amendments for the authors to address prior to this paper being suitable for publication. A list of these are as follows:

1. Title: The authors should consider amending the title to read: “Enhancement of the Oral Bioavailability of Ibrutinib Using a Liposil Nanohybrid Delivery System”

2. Author details: the author’s details should be consistent and in accordance with the requirements of the journal.

3. Spacing: there are numerous spaces omitted between words and between words and references in the manuscript. The authors are advised to do a proof-read of the manuscript to correct these.

4. Spelling and grammar: there are a few spelling and grammar errors in the manuscript, as well as missing units in presenting the methodology and data. The authors are advised to do a proof-read of the manuscript to correct these.

5. Introduction, Paragraph 2, Lines 3-4: The statement ‘…Liposomes with a silica nanoparticle coating…” should be referenced.

6. Referencing: there are inconsistencies regarding the in-text referencing. The authors are advised to do a proof-read of the manuscript to amend all in-text references to conform to the requirements of the journal.

7. RP-HPLC methodology: The authors should state if the RP-HPLC method used for quantification of Ibrutinib was validated and should this method be from literature, a reference should be provided.

8. Preparation of ibrutinib nanoliposomes method: The authors of this manuscript have previously published the methodology used for the preparation of ibrutinib nanoliposomes: Ashar F, Hani U, Osmani RAM, Kazim SM, Selvamuthukumar S. Preparation and Optimization of Ibrutinib-Loaded Nanoliposomes Using Response Surface Methodology. Polymers. 2022; 14(18):3886. https://doi.org/10.3390/polym14183886. This should be included as a reference in this section.

9. Equipment details: The details and manufacturer of all equipment used in the preparation and analysis of the formulations in this manuscript should be provided in the methodology section under the respective methods.

10. Low pH stability method: the temperature for this test should be provided.

11. In vitro cytotoxicity study method: the methodology for the culturing of the S180 cells should be provided including the confluency of the culture media prior to the cytotoxicity study. Additionally, the number of sample runs used for the cytotoxicity study should be provided.

12. In-vitro drug release study: Was this study undertaken at pH 1.2 and pH 6.8 to account for transition of the prepared liposils through the gastrointestinal tract? While the exposure of the liposil to pH 2, 4 and 6 has been evaluated in the Low pH stability analysis, the temperature of this study has not been provided and the time period of 30 minutes is not adequate to determine that no release will take place at pH 1.2 and 6.8 in vivo. In vitro drug release data at pH 1.2 and pH 6.8 should therefore be included in the in vitro release study to correlate between the in vitro and in vivo studies.

13. Drug release modelling: The equations for the kinetic models employed (Korsemeyer Peppa's, Higuchi's, first-order, and zero-order) to model the release data are known and should be removed.

14. In vivo study: the method used for preparation of the ibrutinib suspension should be provided.

15. Characterization of ibrutinib liposils: The zeta potential data is present in Table 1, however this has not been mentioned in the manuscript text.

16. Particle imaging: the quality of the images in Figure 1 is very poor and the scale bars are not legible.

17. Entrapment efficiency results: the results for the Entrapment efficiency are not described in-text.

18. FTIR data: references are required for all peak allocations. Additionally, was all the native components of the liposils analyzed in the FTIR study to compare with the FTIR spectra generated for the nanoliposomes and liposils?

19. Differential scanning calorimetry data: references are required for the allocation of the melting point of ibrutinib at 159.12 °C.

20. Kinetic modeling: the maximum R2 value of 0.7224 is not high enough to provide a correlation with the models used and therefore determine the mechanism of drug release. The authors are advised to investigate other kinetic models to ascertain the drug release mechanism of the developed liposils.

21. Figure 3, Figure 4, Figure 5 and Figure 6 should include error bars.

22. Discussion: there is limited to no discussion of the in vitro and in vivo release data in this section.

6. PLOS authors have the option to publish the peer review history of their article (what does this mean?). If published, this will include your full peer review and any attached files.

Reviewer #1: No

Reviewer #2: No

---

## [Author Response · Author response to Decision Letter 0]

15 May 2024

Response to reviewers: The authors would like to thank the reviewers for giving their valuable time to thoughtfully process and review the manuscript. We are grateful for their inputs and suggestions that have improved the overall quality of the manuscript. 

The authors as advised by the reviewers have addressed all the comments that have been asked in the revision. We are hopeful that our responses meet the acceptance criteria of both the reviewers.

---

## [Decision Letter · Decision Letter 1]

24 Jun 2024

PONE-D-24-01636R1Enhancement of Oral Bioavailability of Ibrutinib Using a Liposil Nanohybrid Delivery SystemPLOS ONE

Dear Dr. Selvamuthukumar,

Thank you for submitting your manuscript to PLOS ONE. After careful consideration, we feel that it has merit but does not fully meet PLOS ONE’s publication criteria as it currently stands. Therefore, we invite you to submit a revised version of the manuscript that addresses the points raised during the review process.

ACADEMIC EDITOR: The authors are advised to address the comments and concerns raised by reviewer #2.==============================

We look forward to receiving your revised manuscript.

Kind regards,

Pradeep Kumar, Ph.D.

Academic Editor

PLOS ONE

Journal Requirements:

Reviewers' comments:

Reviewer's Responses to Questions

**Comments to the Author**

1. If the authors have adequately addressed your comments raised in a previous round of review and you feel that this manuscript is now acceptable for publication, you may indicate that here to bypass the “Comments to the Author” section, enter your conflict of interest statement in the “Confidential to Editor” section, and submit your "Accept" recommendation.

Reviewer #1: All comments have been addressed

Reviewer #2: (No Response)

2. Is the manuscript technically sound, and do the data support the conclusions?

Reviewer #1: Yes

Reviewer #2: Partly

3. Has the statistical analysis been performed appropriately and rigorously? 

Reviewer #1: Yes

Reviewer #2: Yes

4. Have the authors made all data underlying the findings in their manuscript fully available?

Reviewer #1: Yes

Reviewer #2: Yes

5. Is the manuscript presented in an intelligible fashion and written in standard English?

Reviewer #1: Yes

Reviewer #2: Yes

6. Review Comments to the Author

Reviewer #1: (No Response)

Reviewer #2: The authors have significantly improved the previously submitted manuscript and have addressed all comments provided. A few of the amendments however require further expansion before this manuscript can be accepted for publication. A list of these are as follows:

1. The in vitro release data provided in Table 2 should be included in the drug release profile in Figure 6b, accounting for residence time in the gastrointestinal tract at pH 1.2, 6.8 and 7.4. This will allow for a better correlation between the in vitro and in vivo data presented.

2. The referencing of the FTIR data is not complete with the peak designations assigned to the nanoliposomes not referenced.

3. The number of sample runs should be included in the Figure captions. Additionally, where error bars have not been included in the images, the standard deviation should be stated in the Figure caption.

7. PLOS authors have the option to publish the peer review history of their article (what does this mean?). If published, this will include your full peer review and any attached files.

Reviewer #1: No

Reviewer #2: No

---

## [Author Response · Author response to Decision Letter 1]

9 Aug 2024

We believe these revisions address the reviewers’ comments thoroughly and enhance the overall quality and clarity of the manuscript. Thank you for your insightful feedback.

---

## [Decision Letter · Decision Letter 2]

28 Aug 2024

PONE-D-24-01636R2Enhancement of Oral Bioavailability of Ibrutinib Using a Liposil Nanohybrid Delivery SystemPLOS ONE

Dear Dr. Selvamuthukumar,

Thank you for submitting your manuscript to PLOS ONE. After careful consideration, we feel that it has merit but does not fully meet PLOS ONE’s publication criteria as it currently stands. Therefore, we invite you to submit a revised version of the manuscript that addresses the points raised during the review process.

We look forward to receiving your revised manuscript.

Kind regards,

Pradeep Kumar, Ph.D.

Academic Editor

PLOS ONE

Journal Requirements:

**Additional Editor Comments:**

1. The figure caption for Figure 6 should be amended to reflect Figures 6c and 6d. Additionally the label for Figure D in Figure 6 is not legible.

2. The number of sample runs should be included in the figure captions.

Reviewers' comments:

Reviewer's Responses to Questions

**Comments to the Author**

1. If the authors have adequately addressed your comments raised in a previous round of review and you feel that this manuscript is now acceptable for publication, you may indicate that here to bypass the “Comments to the Author” section, enter your conflict of interest statement in the “Confidential to Editor” section, and submit your "Accept" recommendation.

Reviewer #2: All comments have been addressed

2. Is the manuscript technically sound, and do the data support the conclusions?

Reviewer #2: Yes

3. Has the statistical analysis been performed appropriately and rigorously? 

Reviewer #2: Yes

4. Have the authors made all data underlying the findings in their manuscript fully available?

Reviewer #2: Yes

5. Is the manuscript presented in an intelligible fashion and written in standard English?

Reviewer #2: Yes

6. Review Comments to the Author

Reviewer #2: The authors have addressed all comments provided previously. Minor revisions to Figure 6 and the figure captions are required prior to this paper being accepted for publication. A list of corrections is as follows:

1. The figure caption for Figure 6 should be amended to reflect Figures 6c and 6d. Additionally the label for Figure D in Figure 6 is not legible.

2. The number of sample runs should be included in the figure captions.

7. PLOS authors have the option to publish the peer review history of their article (what does this mean?). If published, this will include your full peer review and any attached files.

Reviewer #2: No

---

## [Author Response · Author response to Decision Letter 2]

31 Aug 2024

We appreciate the time and effort put by the reviewers in providing a fair and comprehensive review of the manuscript. In this letter, we have addressed the comments sent to us and hope that we have answered it as per the standards of the Journal

---

## [Editor Report · Decision Letter 3]

2 Sep 2024

Enhancement of Oral Bioavailability of Ibrutinib Using a Liposil Nanohybrid Delivery System

PONE-D-24-01636R3

Dear Dr. Selvamuthukumar,

We’re pleased to inform you that your manuscript has been judged scientifically suitable for publication and will be formally accepted for publication once it meets all outstanding technical requirements.

Kind regards,

Pradeep Kumar, Ph.D.

Academic Editor

PLOS ONE
---

## [Editor Report · Acceptance letter]

13 Sep 2024

PONE-D-24-01636R3 

PLOS ONE

Dear Dr. Selvamuthukumar, 

I'm pleased to inform you that your manuscript has been deemed suitable for publication in PLOS ONE. Congratulations! Your manuscript is now being handed over to our production team.

Kind regards, 

on behalf of

Prof. Pradeep Kumar 

Academic Editor

PLOS ONE